# Simplified Markerless Stride Detection Pipeline (sMaSDP) for Surface EMG Segmentation

**DOI:** 10.3390/s23094340

**Published:** 2023-04-27

**Authors:** Rafael Castro Aguiar, Edward Jero Sam Jeeva Raj, Samit Chakrabarty

**Affiliations:** 1School of Biomedical Sciences, Faculty of Biological Sciences, University of Leeds, Leeds LS2 9JT, UK; 2School of Computer Science Engineering, Vellore Institute of Technology, Vellore 632014, India

**Keywords:** gait detection algorithms, EMG, IMU, segmentation, muscle activity

## Abstract

To diagnose mobility impairments and select appropriate physiotherapy, gait assessment studies are often recommended. These studies are usually conducted in confined clinical settings, which may feel foreign to a subject and affect their motivation, coordination, and overall mobility. Conducting gait studies in unconstrained natural settings instead, such as the subject’s Activities of Daily Life (ADL), could provide a more accurate assessment. To appropriately diagnose gait deficiencies, muscle activity should be recorded in parallel with typical kinematic studies. To achieve this, Electromyography (EMG) and kinematic are collected synchronously. Our protocol sMaSDP introduces a simplified markerless gait event detection pipeline for the segmentation of EMG signals via Inertial Measurement Unit (IMU) data, based on a publicly available dataset. This methodology intends to provide a simple, detailed sequence of processing steps for gait event detection via IMU and EMG, and serves as tutorial for beginners in unconstrained gait assessment studies. In an unconstrained gait experiment, 10 healthy subjects walk through a course designed to mimic everyday walking, with their kinematic and EMG data recorded, for a total of 20 trials. Five different walking modalities, such as level walking, ramp up/down, and staircase up/down are included. By segmenting and filtering the data, we generate an algorithm that detects heel-strike events, using a single IMU, and isolates EMG activity of gait cycles. Applicable to different datasets, sMaSDP was tested in healthy gait and gait data of Parkinson’s Disease (PD) patients. Using sMaSDP, we extracted muscle activity in healthy walking and identified heel-strike events in PD patient data. The algorithm parameters, such as expected velocity and cadence, are adjustable and can further improve the detection accuracy, and our emphasis on the wearable technologies makes this solution ideal for ADL gait studies.

## 1. Introduction

Investigation of gait patterns is a valuable diagnostic tool that provides clinicians with the ability to diagnose a range of impairments affecting a patient’s mobility, such as trauma, stroke, or neurological conditions. Features extracted from these gait patterns, such as key-event times, duration, and joint behaviour properties, are all quantifiable measures that allow a clinician to benchmark and make informed decisions regarding the selection of appropriate therapies.

The core element of gait assessment studies is the identification and analysis of gait cycles. A gait cycle is comprised of two core phases for each limb: stance and swing. This phase differentiation is associated with foot contact with the floor (stance) and swinging motion of the limb through the air, preceding the following step (swing). Taking the example of a right leg dominant subject, Figure 1 describes the gait cycle for a naturalistic walking pattern. For more detailed explanations of the gait cycle, please refer to [1,2,3].

To record and study gait cycles, typical studies rely primarily and solely on the investigation of biomechanics of a subject’s walking behaviours. Kinematic and kinetic analysis of gait allows for the investigation of joint displacements, torques, motor behaviors, and gait patterns, among others. However, alone, it is insufficient to fully characterize a subject’s gait parameters. A purely anatomical/biomechanical exploration of gait disregards important physiological aspects that may be affecting the pathological gait. To mitigate this issue, any biomechanical studies of gait should be accompanied by synchronized muscle activity recordings. Through the association of kinematic analysis with electromyography, a more complete anatomical and physiological investigation of gait is made possible.

Clinical gait assessment relies on a trained clinician’s visual and qualitative assessment of a subject’s biomechanics. Several technologies have been developed over the past decades to assist the clinicians visual inspection and validation of the subject’s mobility patterns, joint angles and general gait analysis. Employed solutions for gait analysis range nowadays from optical recording systems, to mechanical sensors and more advanced artificial intelligence characterisation tools via physiological signals.

The common practice for conducting gait analysis is to capture a subject’s walking behaviour through optical motion capture systems. These systems are highly precise and can capture motion data with millimetre-level accuracy. However, they are restricted to laboratory settings due to the requirement for a controlled environment and equipment setup, limiting the ability to analyse natural walking patterns in real-life situations. Furthermore, the use of these systems is expensive, time-consuming, and requires expert technical staff to set up and operate the equipment. To address these challenges, wearable sensor technology has emerged as a cost-effective and accessible alternative for conducting gait analysis. These sensors can be attached to the subject’s body or clothing and capture kinematic and physiological data during natural walking in unconstrained settings. The use of wearable technology also provides opportunities to conduct long-term monitoring of gait patterns, which can be used to assess changes in a patient’s gait over time and evaluate the efficacy of treatment plans.

To identify the most appropriate sensors for gait event detection during Activities of Daily Living (ADLs) and to pair with EMG analysis, a short review of available sensor technologies was conducted. One of the earliest reported wearable technologies for gait event detection uses foot-switches [4]. These sensors are usually embedded into the shoe or insole, located underneath the heel and hallux of the individual, allowing for the recording of two key gait events: Initial-Contact (IC), reported in healthy scenarios as the heel strike (HS), and End-of-Contact (EC) or Terminal Contact (TC), reported in healthy scenarios as the toe-off (TO). With current iterations such as insole pressure sensors (IPS) and force-sensitive resistors (FSR) fitted into footwear (e.g., [5,6,7]), this technology is widely employed today and used as a benchmark for other motion capture systems (see [8] for an up-to-date review). A common issue with these IPS and FSR systems is the constant wear-and-tear of the sensors due to associated shear forces in the sole or against the walking surface. This leads to short sensor lifespans and sensing accuracy loss after sustained periods of usage.

Nowadays, the most commonly employed wearable technology for gait event detection is accelerometry [8]. First described in [9], this method makes use of acceleration sensors, strategically placed near a subject’s joints on predetermined sites. The choice of placement will greatly influence the gait parameters extracted and takes into consideration possible noise motion artefacts that contaminate the recordings (e.g., poor sensor adhesion or loose clothing). With advances in wearable technology, typical accelerometer sensors were adapted into IMUs (Inertial Measurement Units), sensors that are able to detect both acceleration and angular velocity (through an integrated gyroscope) in all three dimensions. Depending on the desired gait parameters, the IMU sensors are most often placed at the foot [5,10,11,12,13,14,15,16,17,18,19,20] or shank/calf [3,6,10,11,17,21,22], but other studies present interesting alternatives as knee/thigh [5,21,23,24,25], waist [26] or even head [27,28] placements.

During the aforementioned gait events (i.e., HS and TO), there are distinguishable peaks in the vertical acceleration profiles that can be used to identify the timing of HS and TO. Upon contacting the surface, the heel experiences a vertical force (ground-reaction force) and its respective upward acceleration, which is identifiable when inspecting the recorded signal. In addition to gait event detection, IMUs also present benefits in the detection of angular displacements and velocities of different joints [6,15,20] as well as pedestrian tracking solutions [13]. With state-of-the-art sensor synchronisation, current IMU solutions also provide accurate position estimation for the subjects wearing the sensors, as described in [29]. Due to all these features, comparison studies [5,6,7,8,11,20] often place the IMU as a preferable wearable solution over the previously discussed alternatives (i.e., FSR and Optical systems). Even if the IMU is not as accurate as the FSR or as thorough in limb position detection through optical systems, the IMU sensor’s durability, robustness, cost, and applicability outweigh the benefits of these other technologies in ADL gait detection applications.

So far, the wearable technologies presented here rely on external interactions of the individual with the sensors (e.g., pressure applied on the IPS) or measurements of external inferred parameters (e.g., the body part acceleration or angular velocity as measured by the IMU) to identify key gait events. An alternative is to make use of the aforementioned patient’s physiological signals, as muscle activity recordings through surface Electromyography (sEMG), to estimate the timing of the events. sEMG provides a reliable non-invasive solution to analyse muscle activity levels during varied motor tasks, and in a non-invasive manner. sEMG can be used to diagnose motor activity, where the recorded muscle activity is analysed to identify muscle deficiencies, and/or establish individual muscle function levels during pre-defined tasks. This has been specifically used for the analysis of gait, using analysis of the sEMG recorded from hip, knee and ankle flexor and extensor muscles, reported in [2,7,24,25,30,31] with positive evidence for differentiating activity in different segments of the gait cycle. However, as it is a biological signal, sEMG is prone to high noise and variability, depending on sensor location and other biological factors.

To interpret muscle activity features and classify these into gait events, studies often return to the use of machine learning algorithms [7,24,25,31]. Unfortunately, as mentioned, there are several factors that may interfere with the sEMG quality and the accuracy of these classification algorithms, especially when recognising how volatile and different the recorded signals can be across different individuals, and even in different muscles of the same individual.

All these wearable solutions present promising results for gait event detection, through varied technologies and algorithms. The complexity of such tools decides how viable it is for applications of gait assessment during ADLs. A summary of the advantages and disadvantages of the discussed technologies is presented in Table 1. Unfortunately, these solutions seldom come with a clear methodology of how to apply these tools to detect the most simple of gait events, rarely offering the algorithms to do so. In this method paper, we propose a simplified method of heel strike detection for synchronised sEMG segmentation through IMU data paired with sEMG recordings, applied to the diverse and unconstrained dataset of [32]. This method allows for a joint biomechanical and physiological investigation of gait. We further test sMaSDP with kinematic IMU samples of data from [33], looking at Parkinson’s Disease (PD) gait.

## 2. Methodology

The following section details the steps, and associated scripts, to isolate sEMG activity patterns in different unconstrained gait modalities, based on the kinematic dataset of [32]. This guide contains step-by-step information about using recordings from an IMU system [34], with focus on an IMU sensor on the subject’s dominant leg foot, to identify heel strike events and timestamps during different gait modalities. The presented pre-processing and processing steps described for the gait event detection and sEMG segmentation are demonstrated and discussed in the following separate subsections. The designed presented algorithms were coded in *MATLAB* [35].

The dataset provided in [32] contains unconstrained walking data from 10 healthy subjects (five males, five females, ages between 18–31 years old), with no known neurological disorders. Kinematic (through IMU system [34]) and sEMG recordings were collected from each subject participating in the experiment. For the design of the sMaSDP algorithm herein presented, special focus was given only to the sensors presented in Figure 2.

In this experiment, subjects walk through a designed walking course, encompassing a ramp, level ground and staircase sections, for a total of 20 trials per subject. In each trial, the subject walks up and down the ramp and staircase, as well as a section of level ground walking, providing recordings for five different walking modalities. For ease of referencing, please consider the following acronyms:RA: ramp ascent.RD: ramp descent.SA: staircase ascent.SD: staircase descent.LGW: level ground walking.

To accompany and summarise the following methodology, Figure 3 details the required steps and processes for the kinematic and sEMG data.

### 2.1. Interpolation and Kinematic Activity Segmentation

Kinematic data (KIN) and sEMG data were recorded at the synchronously, but they have different sampling rates. Temporal synchronisation ensures that both recordings have the same duration, but the number of samples captured for each signal during this time varies due to the different sampling rates. The difference in sampling frequencies between KIN and sEMG is primarily due to the sample capture and processing capabilities of the recording equipment, as well as the amount of data generated per sample. Optical kinematic systems typically use a sampling frequency of around 120 Hz [36], which is much lower than the typical frequencies of 1000 Hz or higher for sEMG systems. For IMUs, research has shown that a capture frequency of 60 Hz is sufficient to record kinematic events while maintaining acceptable processing requirements [37]. However, for sEMG, the higher end of the sEMG signal frequency band can reach 500 Hz (see [38]). Therefore, to capture all the nuances of the sEMG signal effectively, a sampling frequency of 1000 Hz or higher is required according to the Nyquist frequency principle. In the dataset of [32], sEMG was recorded at a sampling frequency of 1000 Hz, while the IMU was recorded at a sampling frequency of 60 Hz. Consequently, sEMG data had approximately 17 times more samples than IMU data.

To use kinematics as an indicator for segmentation timestamps in the sEMG counterpart, the kinematic signal requires upsampling to match the number of sEMG datapoints. This process ensures that both KIN and sEMG recordings are synchronised in time and samples, in order to use the kinematic datapoints to achieve pinpoint accuracy. A moving average smoothing filter is then applied to KIN to remove unwanted oscillations and initial motion noise from the signal. After the smoothing filter is applied, the kinematic signals are upsampled via linear interpolation (Equation (Equation 1)):(1)$$y=y_{1}+\left(x-x_{1}\right)\frac{\left(y_{2}-y_{1}\right)}{\left(x_{2}-x_{1}\right)}$$
where *y* is the interpolated value, *x* is the sample datapoint to perform interpolation, $y_{1}$ and $y_{2}$ are the first and second signal samples, and $x_{1}$ and $x_{2}$ are the first and second datapoints.

This pre-processed sEMG and the upsampled KIN data consist of two complete walk trials, each including all five walking modalities previously discussed (RA, RD, SA, SD, and LWG) for each subject. A minimal energy threshold and window is created to isolate and timestamp meaningful movement identified using the IMU acceleration. Any signal fragment discovered inside a passing window that is less than a certain energy level is categorized as ‘0’. Similarly, if significant acceleration is generated beyond the threshold regarded as evidence of the subject walking, it is recorded as ‘1’. As a result, a binary signal is generated identifying data points of relevant walking activity.

### 2.2. Filtering Small Movement Artefacts and Separating Trials from a Single Recording

The core objectives of this section are to filter out movement and/or transition artifacts from the IMU signal, considered as noise, and to identify the direction of movement, as the walking modalities presented are dependent on the direction of the walking course. The output of the previous section is a duty-cycle type wave, depicted as the grey square wave in Figure 4A, which presents binary values of 1 when the subject is active and 0 when the subject is standing still. The start and end of active periods define the timepoints at which walking starts and ends. As the subjects complete the trials continuously without taking unnecessary breaks between walking modalities, any detected activity below a threshold of 6000 data points (or 6 s) is classified as a motion artifact or noise.

In the experiment, each recording consists of two trials. In a single trial, a subject walks the course in a forward direction, rests at a designated location (B), and then returns to the starting point (A) by walking in the reverse direction of the course. Each walking direction (A-B and B-A) is defined as a “half-trial”. Therefore, a single recording has four half-trials, with two being the subject walking from the starting point (A) to point (B), and two where the subject walks in the reverse direction from point (B) to point (A). From the complete recordings, four moments of activity are extracted and stored as an independent variable to define the direction in which the subject walked the course.

### 2.3. Initial sEMG Segmentation and Direction, Using the Timepoints Defined by the Four Moments of Activity per Complete Trial

The extracted timestamps for the start and end of each half-trial are used to segment the sEMG and KIN signals. The sEMG recordings provided in this dataset are in raw format. The objective of this stage is to remove noise from movement artefacts, possible recording interference, and isolate relevant frequency content from the data. To achieve this purpose, a fourth-order Butterworth bandpass filter is designed to filter out any sEMG signals outside the 10 to 150 Hz band. Additionally, a notch filter is used to remove powerline interference at 60 Hz.

The next objective is to cut both sEMG and KIN signals using the previously defined half-trial timestamps. For this purpose, a timestamp with a safety margin of 2000 data points (2 s) is used to ensure the inclusion of all relevant activity. sEMG and KIN are then segmented into activity bins based on half-trials for all trials and all subjects.

As mentioned, per complete trial, four moments of walking activity are detected (example in Figure 4A), related to walking the course twice in the forward and twice in the reverse directions, as described previously between (A) to (B). Based on the extracted timestamps and the experiment methodology, the sequence of modalities in the forward direction is separated into RD, LGW and SA, and for the reverse course direction, separated into SD, LGW and RA.

### 2.4. Walking Modality Identification within Each Direction of the Course and Trials

At this stage, sEMG and KIN signals for all trials are segmented and separated according to the direction of the course the subject is walking along. The next step is to identify the timestamps for each walking modality within the forward and reverse courses, and isolate these in both the sEMG and KIN signals. Due to the design of the course, subjects take a sharp turn at the transition between each modality (example on Figure 4B). This is identified via the estimated position changes as recorded by the IMU. Focusing on the *y*-axis, any time there is a peak transition, a new modality begins, according to the course guide mentioned earlier. This, paired with safety margins, is then used to further segment both sEMG and KIN into the individual walking modalities.

### 2.5. Using Acceleration Data to Identify the Moment of Heel Strike

Having the data segmented into different walking modalities, the next step is to identify gait cycles within each walking modality. Through acceleration data in the vertical direction (*z*-axis), the moment of HS can be identified. To do so, new filters are designed to isolate the vertical peaks of acceleration witnessed when the heel strikes the ground, and experiences the ground-reaction force. Two separate seventh-order Butterworth filters are created: a high-pass at 9 Hz and a low-pass at 6 Hz. These remove a band where specific motion artefacts lie, and isolate the striking moments. Being post-hoc, these filters are ideal and do not impose any delays on the data.

The filtering process is as follows:the high-pass filter is applied to the raw KIN data, removing frequencies below the 20 Hz band, and potential movement artefacts.a half-wave rectifier is applied to the resulting filtered data, removing unnecessary negative electrical oscillations imposed by the sensor readings.a low-pass filter is applied to the rectified data, serving as an anti-aliasing filter at 5 Hz, to remove oscillations from the signal.

The HS moment is identified as the highlighted peaks in the Filtering and Peak section of Figure 4C. Extracting the timestamps of each of these peaks allows then for the isolation of a complete stride, or gait cycle, from one heel strike moment to the immediate next of the same limb. These timestamps are then finally used to segment the sEMG activity into separate gait cycles.

## 3. Results

Figure 4C presents the identified the moments of HS, for a segment of level ground walking, from a randomly selected subject. In the other walking modalities, there was a noticeable shift in the amplitude of the ground-reaction force acceleration and an adjusted cadence. Through simple adjustment of the expected acceleration peaks allowed for HS to equally as recognisable and identified in these different walking modalities using sMaSDP.

Through the identified gait event timestamps, computed from the IMU data, individual gait cycles are then extracted. Using the timestamps of each gait cycle, sEMG and kinematic data is segmented per cycle. In Figure 5, an example of sEMG activity is presented for the same level ground walking segment of Figure 4. Within the isolated gait cycles, we can identify the muscle activity patterns from the segmented sEMG data, as seen in Figure 6. Both biomechanical data and physiological data can be correlated at this point, to best characterise this subject’s gait. The noticeable variability seen in the muscle activity levels of Figure 6 highlights once more the necessity of a paired kinematic measure associated with the this physiological data to best discriminate the witnessed motions. This muscle activity variability also demonstrates the adaptability of our physiological neural control signal, even in seemingly simple and rhythmic motions, as level ground gait.

As a proof of concept, this algorithm was applied to another unconstrained walking dataset [33], to PD patient data, where subjects performed a series of walking tasks in a therapy scenario. PD patients have known abnormal gait patterns, as irregular stepping and shuffle gait. In the dataset of [33], patients were fitted with an IMU placed on their dominant foot, following a similar placement as seen in [32] and Figure 2. Unlike the dataset [32] used to design our algorithm, the experiment with the PD patients did not collect sEMG recordings, and the focus in this scenario is only on the kinematic gait data. With fine tuning of cadence and velocity parameters, through adjustments of the peak detection section, all the moments of HS are detected over an entire trial (over 1 min). Results for the sMaDSP detection of the HS events, from a randomly selected trial and subject of this dataset, are presented in Figure 7.

## 4. Discussion and Conclusions

Tracking patients’ performance levels at their own homes and through their daily lives is of massive benefit for the design of smarter continued therapies. With the ambition of continued, supervised at-home therapies and improved rehabilitation, it is essential that researchers are provided with simple guides on using the technology and appropriate methodology to provide them the foundation for exploring therapeutic mobility. With advancements in sensor design, it is now possible to bring these tools to our everyday lives, and gait assessment to activities of daily living.

Even in the ADL scenario, as demonstrated, kinematics alone may not provide sufficient information to therapists to fully assess patients’ mobility. Integration of muscle activity levels, associated with recorded movements and gait events will benefit not only diagnosis but performance assessment. With simplification of the acquisition and processing algorithms, as well as selection of appropriate wearable sensors, a platform that integrates both kinematics and sEMG for home use becomes feasible. With this report, a simplified guide for kinematic and sEMG recording segmentation, and gait event detection was presented and fully documented. Codes and algorithms, that work with the dataset of [32] are made available alongside this method. To adapt to other datasets, users can easily adjust expected cadence or filter level variables, improving detection and allowing the study of different walking modalities, providing an initial framework for researchers to explore and adjust to their needs and data.

Synchronising both sEMG and kinematic information is the step forward to better characterise movement, and it is crucial that both signals are analysed concurrently. Analysis of non-invasive sEMG recordings in ADLs should be prioritised to define the patterns of muscle activity that generate movement. The same level of “wearable” exploration that kinematics and gait cycle has received over the past decades, needs to be pursued for the sEMG case.

With adjustable parameters, our sMaSDP algorithm can be adapted to further ADLs scenarios, and potentially applied to patient’s smartphones, for a continuous kinematic recording. Due to its simplicity and focus to aid newcomers to the gait studies area, this algorithm can also be combined with machine learning in future iterations, to allow ad-hoc heel strike identification. 

## Figures and Tables

**Figure 1 sensors-23-04340-f001:**
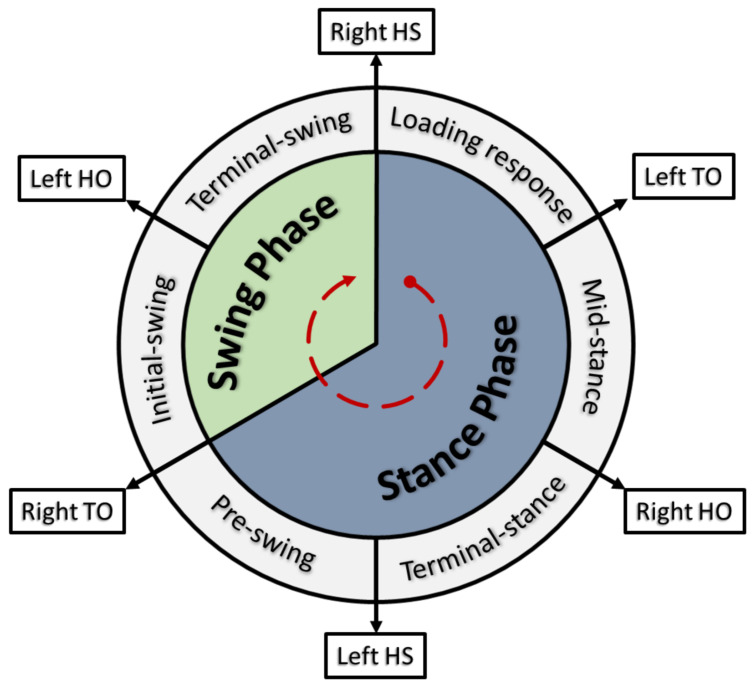
Simplified stages of a typical gait cycle for a right leg dominant subject. A new cycle begins with the right heel strike (HS) and both feet in contact with the floor. After a loading response, the left foot is lifted off the floor following the left Toe-off (TO), and weight starts shifting on the right foot from heel to mid-foot as the body moves forward. Once the body reaches mid-stance, the right heel loses contact with the floor during the right Heel-off (HO) stage, and weight shifts towards the right forefoot. To prepare for the right limb swing phase, the left foot contacts the floor again with a left heel strike (HS), and both feet are in contact with the floor, which is called double support. The right foot then enters the swing phase following the right Toe-off (TO), and the same stance process happens on the left foot, with weight shifting from heel to toe, and the respective left Heel-off (HO). The current gait cycle ends with the right HS, and a new cycle starts.

**Figure 2 sensors-23-04340-f002:**
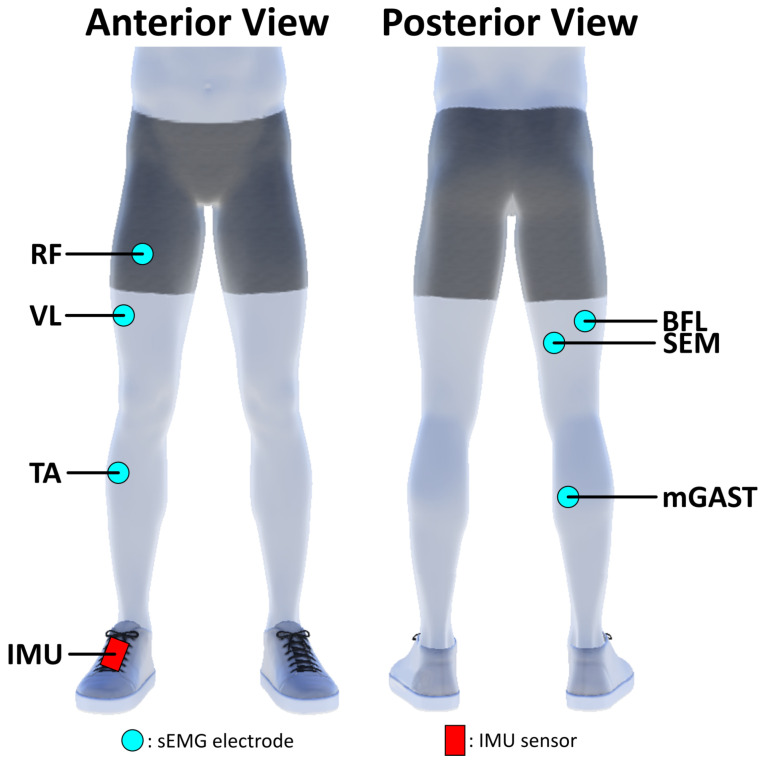
Schematic of the sEMG and IMU sensor placement locations. sEMG recordings were taken from Tibialis Anterior (TA), Medial Gastrocnemius (mGAST), Vastus Lateralis (VL), Rectus Femoris (RF), Semitendinosus (SEM) and Biceps Femoris Longus (BFL). The IMU sensor was attached on top of the subject’s shoe, at the centre of the right foot.

**Figure 3 sensors-23-04340-f003:**
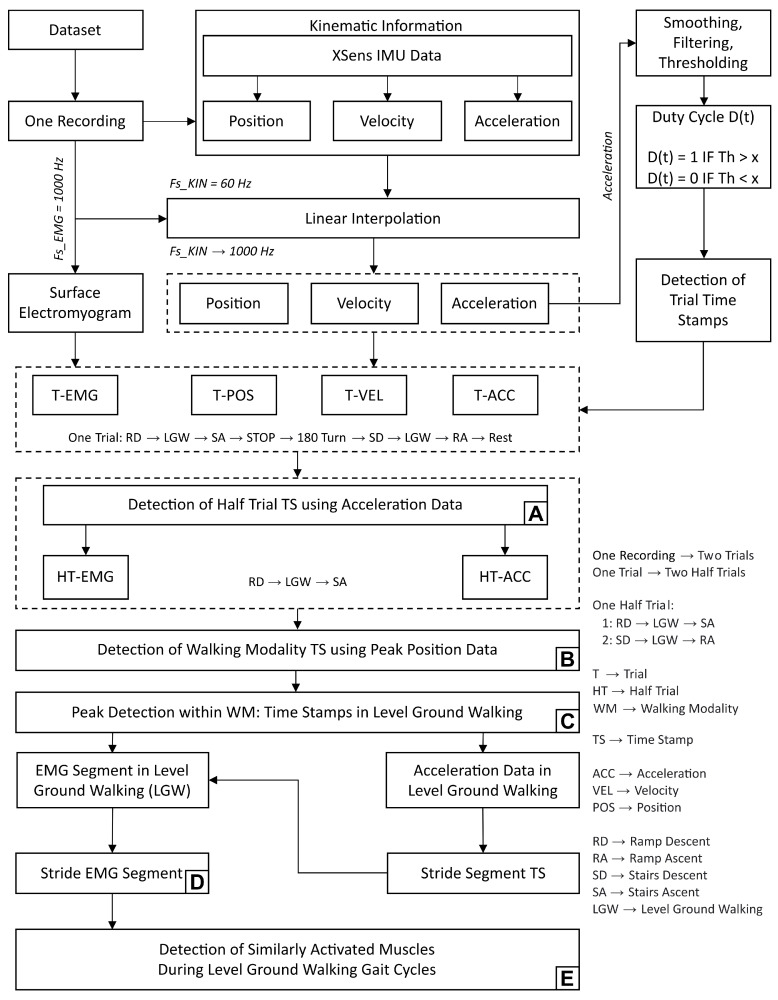
Summary of method architecture: the diverse multi-step process, from the recorded IMU signals, filtering of movement periods, segmentation into different walking modalities and identification of HS moments. Steps labelled **A** to **E** establish a logical link between this architecture and the results.

**Figure 4 sensors-23-04340-f004:**
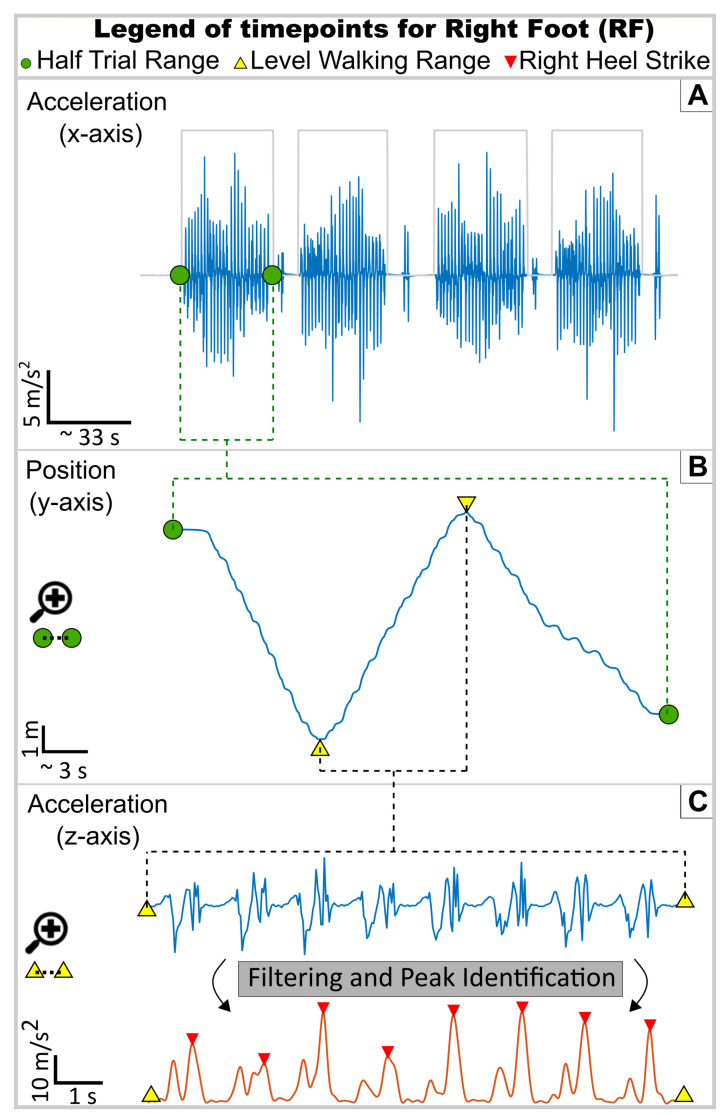
Summary diagram illustrating the different stages of the method, presented and linked in Figure 3, for the duration of an HT (RD, LGW, SA): in Section (**A**), the *x*-axis acceleration of the right-foot IMU is used to identify the initial HT start and end timepoints, marked with the green circles; within the time range defined by the HT green circles, Section (**B**) now takes the *y*-axis position data from the IMU to identify the sharp direction turns related to switches in walking modality, represented here by the yellow triangles; lastly, within the isolated LGW modality between the yellow triangles, Section (**C**) uses the *z*-axis acceleration from the IMU, and through our filtering and peak identification algorithm, identifies the HS events, marked by the red triangles.

**Figure 5 sensors-23-04340-f005:**
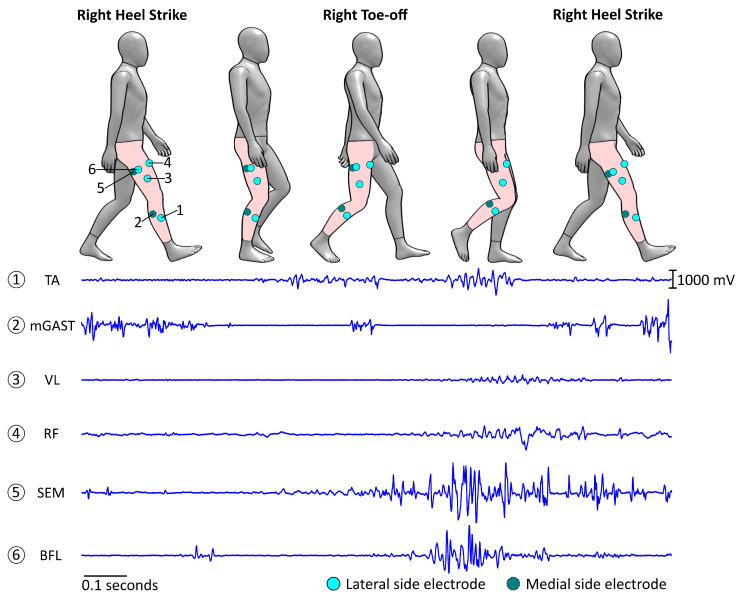
Segmented sEMG from a single extracted gait cycle: top section diagram represents a full gait cycle, as initially described (Figure 1), with the respective locations of the sEMG sensors; the bottom traces are the segmented sEMG recordings for the respective gait cycle. Linked, as labelled, to step **D** of the method architecture (Figure 3).

**Figure 6 sensors-23-04340-f006:**
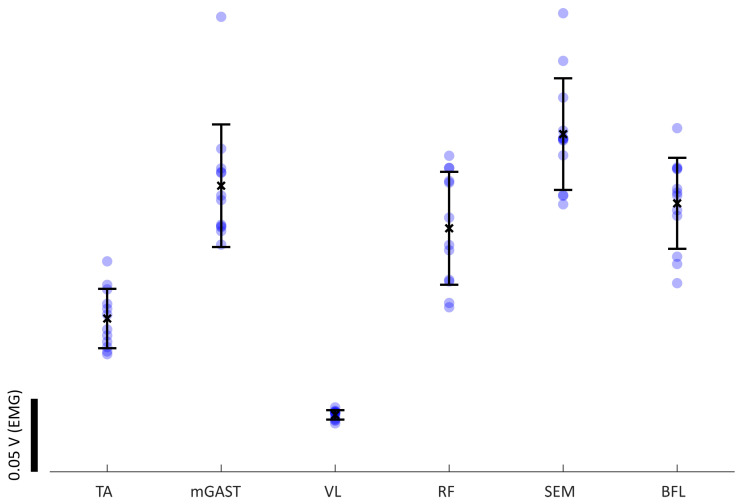
Observations with mean and standard deviation of muscle activity extracted from 12 LGW gait cycles from a randomly selected trial and subject. Analysing this activity in different walking modalities allows for quantification of the variability in each muscle and the impact on the person’s mobility. Linked, as labelled, to step **E** of the method architecture (Figure 3).

**Figure 7 sensors-23-04340-f007:**
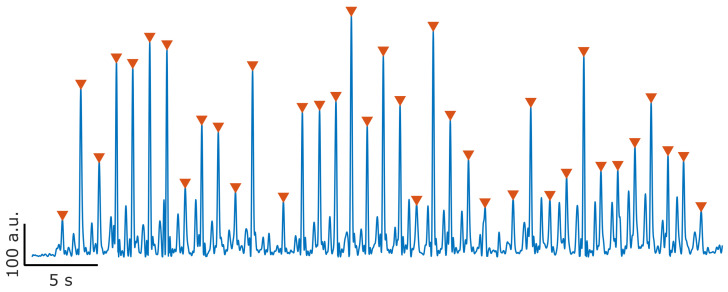
HS detection on unconstrained Parkinson gait, computed from IMU data from the publicly available dataset of [33,39]. The presented data were recorded from a PD patient, supervised by a therapist. Within this recording the PD patient walks freely in three separate modalities: straight walking, “random walking” and ADL walking. HS events are here detected for all modalities.

**Table 1 sensors-23-04340-t001:** Summary of advantages and disadvantages of presented gait event detection solutions, in terms of applicability, interface, robustness and cost.

Sensor Type	Advantages	Disadvantages
Optical Systems	Full-body kinematics. High marker detection precision and accuracy. Accurate estimation of joint position, angle, velocity and acceleration. Synchronisation solutions with other gait assessment hardware (EMG kits).	High cost and maintenance. High performance requirements. Restricted experimental environments. Multiple camera synchronisation and calibration. Setup and collection learning curve. Imposes long donning and doffing times of retroreflective markers, as well as initial setup per subject.
FSRs and IPSs	Ideal for detection of key gait events as heel-strike and toe-off. Low-cost solution, with lower number of sensors required for event detection. Typical benchmark for other gait assessment technologies	Low durability, sensor wearing out. Depending on sensor application site, may impact the subject’s walking style. Uneven terrain and uncharacteristic gait behaviours may lead to accidental noise event recordings.
IMUs	Simple configuration and multitude of solutions to interface with other hardware (EMG kits). Low-cost and high availability. Applicability to diverse gait environments, especially when using wearable solutions. Easy and quick donning and doffing of the sensors to and across subjects.	Lower accuracy. Wearable solutions are battery dependent. High susceptibility to motion artefact noises, depending on application site and surface (directly on the skin or over clothing). Position estimation requires multiple synchronised sensors.
Surface EMG	Valuable physiological information of muscle activity patterns involved in gait. May be used to identify personal muscle recruitment schemes used per individual.	Extremely high inter-subject variability. Hard to characterise gait by itself and requires significant subject by subject adjustments.

## Data Availability

Algorithms developed in this study are openly available in Castro Aguiar, R.; et al. Simplified markerless stride detection pipeline (sMaSDP) for surface EMG segmentation (v1.0). https://doi.org/10.5281/zenodo.7628979 (accessed on 15 May 2022). The data used in this study are openly available in Brantley, J.; et al. Figshare. https://doi.org/10.6084/m9.figshare.5616109.v5 (accessed on 5 October 2020) and Bachlin, M.; et al. https://archive.ics.uci.edu/ml/datasets/Daphnet+Freezing+of+Gait (accessed on 15 May 2022).

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
