# Peer review of "Simplified Markerless Stride Detection Pipeline (sMaSDP) for Surface EMG Segmentation"

_sensors, 2023, doi:10.3390/s23094340_

Round 1
Reviewer 1 Report
This paper applied IMU data to match EMG signals. The idea is novel, but the applicability of the algorithm was not clearly introduced. It is suggested to modify from the following aspects:
1. It was suggested to add the position diagram of the IMU sensors.
2. As Section 2.1, Why not use the same sampling rate to obtain two kinds of data for interpolation?
3. What signals were filtered in Section 2.2?
4. There are no specific parameters about the database, including sample size or the number of experimentalists.
5. In the third part, the single IMU signal has been able to better identify gait, so why the EMG signal was superimposed? If there was improvement, specific indicators should be added for comparison.
6. Line 233 of the third part, the data set used in the experiment did not contain EMG signals, so what’s significance of the experiment?
7. In the conclusion part, no specific parameter index was given to verify the advantages of the proposed algorithm and no comparison was made with other algorithms.
The engineering significance of this paper was not clear either, so it was suggested to clarify the engineering significance.
Author Response
Dear Reviewer,
Annexed is the letter answering your queries.
Kind regards,
Rafael Aguiar

Reviewer 2 Report
The manuscript provides protocol of stride detection pipeline (sMaSDP) for surface EMG segmentation. This article is important from a theoretical point of view for professionals who work with the patients with gait disorders. Nevertheless, it is necessary to make a few comments, the work on which, in my opinion, will help improve the quality of your manuscript.
1. Could you comment equations 1 and 2?
2. You should give a link on Figure 6 in the text of your manuscript.
3. It remains unclear the aim of EEG recording (p.4 line 122).
Author Response
Dear Reviewer,
Annexed I am submitting a letter answering your queries.
Kind regards,
Rafael Aguiar

Round 2
Reviewer 1 Report
-
The content of the article should be revised as appropriate.